# Molecular Basis of Persister Awakening and Lag-Phase Recovery in *Escherichia coli* After Antibiotic Exposure

**DOI:** 10.3390/ijms27010467

**Published:** 2026-01-01

**Authors:** Karolina Stojowska-Swędrzyńska, Ewa Laskowska, Dorota Kuczyńska-Wiśnik

**Affiliations:** Department of General and Medical Biochemistry, Faculty of Biology, University of Gdańsk, Wita Stwosza 59, 80-308 Gdańsk, Poland; ewa.laskowska@ug.edu.pl (E.L.); dorota.kuczynska-wisnik@ug.edu.pl (D.K.-W.)

**Keywords:** aggregates, ATP, β-lactams, DNA repair, *Escherichia coli*, fluoroquinolones, lag phase, persister cells, stringent response, toxin–antitoxin system

## Abstract

Antibiotic persistence is a transient phenotype in which a subset of genetically susceptible bacteria survives lethal antibiotic exposure without acquiring resistance. However, survival alone does not define a persister cell—only cells that successfully recover, resume growth, and produce viable progeny complete the persister cycle. Recent studies in *Escherichia coli* show that persister awakening is a multistage process shaped by dormancy depth, metabolic state, and antibiotic-induced damage. Upstream induction mechanisms, including stringent-response signaling and toxin–antitoxin–mediated growth arrest, primarily determine dormancy depth but do not directly control awakening kinetics. During the lag phase, persister cells undergo coordinated recovery involving detoxification of residual antibiotics, ATP restoration, dissolution of protein aggregates, and ribosome reactivation. After exposure to fluoroquinolones, awakening additionally requires SOS-driven DNA repair via homologous recombination or transcription-coupled repair. In contrast, β-lactam–exposed persister cells rely mainly on efflux-mediated detoxification and asymmetric damage partitioning. Failure to restore proteostasis or resolve damage results in abortive recovery or cell death. Only after damage processing and metabolic reactivation can persister cells resume division and generate viable progeny. This review integrates current molecular insights into persister cell recovery in *E. coli*, highlighting the lag phase as the critical barrier between survival and true persistence.

## 1. Introduction

### Fundamental Concepts, Clinical Relevance of Persistence, and the Role of Stochasticity and Stress in Dormancy Induction

Bacteria have evolved multiple strategies to endure antibiotic exposure. The most widely recognized mechanism is genetic resistance, which enables bacterial growth in the continued presence of antimicrobial agents [1,2]. In contrast, persistence is a non-heritable phenomenon in which a small sub-population of cells transiently tolerates otherwise lethal antibiotic concentrations [3]. Persister cells represent phenotypic variants within an isogenic population that remain viable despite exposure to antimicrobials. They typically constitute only a minor fraction of the population, display little or no growth during treatment, and return to a drug-susceptible state once the antibiotic is removed. Importantly, persistence does not involve an increase in the Minimum Inhibitory Concentration (MIC) [3,4,5].

Unlike resistant mutants, persister cells do not replicate in the presence of antibiotics but evade killing by entering a transient, drug-tolerant physiological state [3]. This distinguishes persistence from heteroresistance, where a genetically identical subpopulation exhibits a markedly elevated MIC, and from tolerance, which describes a non-genetic increase in survival time under antibiotic treatment without affecting MIC [6].

Importantly, heteroresistant cells differ from persisters not only in their ability to grow at antibiotic concentrations above the population MIC, but also in the reversible nature of this phenotype [7]. Heteroresistance can arise through transient, non-genetic regulatory shifts as well as unstable resistance determinants [8]. For example, heteroresistant *Escherichia coli* can rapidly transition between highly resistant and susceptible states once antibiotic pressure is removed, reflecting an active, reversible adaptive response rather than the growth arrest characteristic of persister cells [9,10]. This also contrasts with antibiotic tolerance, which describes a non-genetic delay in killing without affecting MIC, whereas heteroresistance, by definition, involves subpopulations with elevated MIC values.

Clinically, persister cells contribute to the chronic and relapsing bacterial infections [11]. They have been implicated in diseases such as *Mycobacterium tuberculosis* infection [12,13,14], cystic fibrosis–associated lung disease [15], urinary tract infections [16,17], and Lyme disease [4,18]. Their presence is particularly pronounced in biofilms, where they support persistent infections and survival in vivo [19]. Persister cells are also associated with intracellular survival within macrophages [20] and may contribute to immune evasion and treatment failure, especially in immunocompromised hosts [21].

Furthermore, persistence may serve as a stepping stone toward the evolution of genetic resistance. By allowing for the survival of a viable subpopulation during antibiotic exposure, persister cells increase the likelihood of acquiring resistance-conferring mutations, facilitating the long-term emergence of antibiotic resistance within bacterial communities [22].

Studies on bacterial persister cells show that persistence does not arise from a single pathway. It is a result of both responsive and stochastic processes. In the responsive model, environmental cues such as nutrient limitation, oxidative or acidic stress, DNA damage, activation of toxin–antitoxin systems, or exposure to sublethal antibiotic concentrations can trigger a transition into a persistent state, a phenomenon termed a “responsive switch” [23,24]. These signals frequently converge on pathways that suppress cellular activity by inducing translational arrest, lowering ATP levels, or reducing proton motive force [5,25,26,27]. Through metabolic downregulation, cells become transiently tolerant to antibiotics whose efficacy depends on active cellular processes [28,29].

In contrast, the stochastic model views persistence as arising from random fluctuations in gene expression or metabolic activity that transiently halt growth. This mechanism is considered a form of bet-hedging, in which a minority of slow-growing or non-growing cells acts as a safeguard against sudden stress, increasing population survival probability [6,30]. As a result, persister cell populations are heterogeneous, exhibiting differences in metabolic activity, growth rate, morphology, and stress responses [5,31,32].

For many years, persister cells were characterized as dormant cells that survive antibiotic exposure due to the inactivity of essential processes and lack of target engagement [4,21]. Time-lapse microscopy confirmed that growth-arrested cells are enriched among surviving cells; however, only a subset of these survivors successfully resumes growth and fulfills the definition of persister cells [33].

Additionally, stationary-phase cultures, which contain a higher proportion of non-dividing cells, are enriched in persister cells compared to exponentially growing populations [5,34,35]. However, dormancy alone is not sufficient, as only a subset of non-growing cells withstands antibiotic treatment [36].

Importantly, persister cells can also emerge from actively dividing cells. Using cell division reporters and FACS analysis, Orman and Brynildsen demonstrated that while most persister cells originated from non-growing subpopulations (~1%), approximately 20% arose from rapidly dividing cells [33]. Similar observations from microfluidic studies showed that *E. coli* persister cells tolerant to fluoroquinolones and β-lactams were dividing normally before antibiotic exposure [36,37]. These findings indicate that growth arrest is not required for persistence; both dormant and proliferating cells can enter the persister state, emphasizing phenotypic heterogeneity over a single deterministic pathway [37].

Overall, persister cell formation likely results from the interplay between stochastic fluctuations and stress-induced responses. Despite diverse triggers, only a small fraction of cells enters the persister state. Reduced metabolic activity and energy limitation are central features of this transition, although the precise molecular drivers remain to be fully defined [25,38,39].

Historically, research has focused on mechanisms that generate persister cells [3,4,5,40,41]. More recent work has shifted toward understanding awakening and regrowth [42,43,44,45]. Complete insight into persistence requires studying both entry into and exit from the persister state [28].

This review focuses on the post-antibiotic recovery of *E. coli* persister cells, emphasizing the lag phase as a critical window for damage processing. During this period, detoxification, metabolic reactivation, proteostasis restoration, ribosome reawakening, and, when required, DNA repair determine whether growth can resume. *E. coli* serves as the primary model due to extensive characterization and the availability of high-resolution single-cell data. Mechanistic insights mainly derive from studies using β-lactam and fluoroquinolone antibiotics. To ensure clarity, key terminology is summarized in Table 1. This review highlights awakening as the defining phase of true persistence.

Literature search and article selection. This narrative review is based on a focused survey of the literature identified using PubMed and Web of Science. Priority was given to recent publications whenever possible; however, earlier seminal studies were included when they provided foundational mechanistic insights or represented the first characterization of key processes. Article selection emphasized studies that employed genetic perturbations, single-cell imaging, and microfluidic approaches to enabled direct analysis of persister cell physiology and recovery dynamics. The literature survey was completed in September 2025.

## 2. Heterogeneity in Persister Awakening Dynamics

### Diverse Recovery Timings Reveal Graded Survival Outcomes After Antibiotic Exposure

Once antibiotics are removed and fresh nutrients are provided, surviving cells may resume growth. Recovery and regrowth are essential for recolonization of the environment, and thus represent a critical component of persistence. However, awakening from dormancy is inherently asynchronous. Single-cell studies have demonstrated that the timing of regrowth following antibiotic removal varies widely among surviving cells [5,46,47]. Some persister cells resume division almost immediately after drug withdrawal (Figure 1a), whereas others require several hours (up to <12 h) before initiating the first division (Figure 1b), consistent with classical descriptions of persister cells defined by delayed regrowth after antibiotic removal [37,47]. A distinctly delayed group of survivors only restarts division after prolonged dormancy (12–40 h) (Figure 1c) [37]. These “late regrowers” contribute to long-tail survival observed in population-level killing curves, although they may be underrepresented in single-cell microfluidics due to rapid overgrowth by “early regrowers” [37].

In contrast, another subset of surviving cells remains non-dividing for extended periods, even days after antibiotic removal (Figure 1d). These cells are believed to enter a deeper dormant state, frequently referred to as the viable but nonculturable (VBNC) state [34,46,49,50]. VBNC cells typically maintain membrane integrity and may exhibit residual metabolic activity, allowing for distinction from dead cells by propidium iodide staining [50]. However, recent work shows that many VBNC cells are structurally compromised “cell shells” lacking essential cellular components (including DNA) and are therefore unable to resuscitate despite intact membranes [48,51]. These cells represent terminally damaged survivors rather than reversible persister cells [48,52,53].

Once a persister cell completes its first division, it immediately resumes exponential growth, showing doubling times similar to antibiotic-sensitive exponential-phase cells [45,47]. This confirms that persistence is transient and does not impose long-term fitness costs on progeny [35,47]. Thus, persistence is defined not only by survival during treatment but also by the ability and success of awakening [35].

The time required for a persister cell to resume division—commonly referred to as lag time—represents a key transition phase in which the cell shifts from dormancy to an antibiotic-sensitive, growth-competent state. Even under identical awakening conditions, lag duration varies considerably among cells. This variation reflects a type of physiological “memory” shaped by the metabolic and structural state at dormancy entry. According to the “last-in, first-out” model proposed by Jõers and Tenson, cells that entered dormancy most recently awaken first, while deeply dormant cells require extended time to reactivate [34].

During the lag phase, persister cells reverse dormancy-associated adaptations, including restoration of proteostasis, reactivation of ribosomal and replication machineries, and dissolution of protein aggregates accumulated during energy depletion [50,54]. Cells closer to a pre-dormant physiological state awaken faster, whereas deeply dormant cells face greater metabolic and regulatory barriers [34,35,47].

Consequently, dormancy exists on a continuum rather than as a binary state [35]. Cells in shallow dormancy resume growth rapidly after antibiotic removal. Intermediate cells require partial metabolic reactivation. Deeply dormant cells show long delays and many fail to resume growth. At the end of this continuum lie VBNC or failed persister cells, which remain metabolically impaired and cannot divide. Dormancy depth, therefore, determines the likelihood of awakening and the variability in lag-time [34,35,50].

## 3. Nutrient-Sensing and Signal-Driven Control of Resuscitation

### Persister Awakening Is Actively Triggered by Environmental Cues Rather than Random Activation

The heterogeneous and delayed resumption of growth in persister cells was initially attributed to stochastic processes [55] and interpreted as a form of bet-hedging [34]. According to this model, individual cells within an isogenic population resume growth at different times due to random fluctuations in molecular states. This temporal variability may function as an evolutionary insurance strategy, enabling a fraction of cells to exploit favorable conditions immediately, while others remain dormant to withstand potential secondary stresses [55].

However, emerging evidence indicates that persister cell awakening is not purely stochastic but instead constitutes a regulated and signal-driven physiological process [54]. The observation that a subset of persister cells resumes growth almost immediately after antibiotic removal, following exposure to favorable environmental cues, suggests an active capacity to sense environmental improvement [47]. Jõers and Tenson showed that recovery kinetics depend on the available carbon source: glucose promotes rapid regrowth whereas gluconate delays awakening [34]. This indicates that persister cells evaluate nutrient quality and adjust awakening timing accordingly. Glucose, therefore, acts not only as a metabolic substrate but also as a positive environmental signal indicating favorable growth conditions [34].

Persister cell resuscitation is now understood to be triggered by nutrient sensing. Specific metabolites such as glucose [34,54] or alanine [54] initiate awakening through detection by the phosphotransferase system (PTS) and chemotaxis signaling machinery, converging on the regulation of intracellular cAMP [54]. During glucose-induced resuscitation, the PTS component PtsG facilitates sugar import, thereby dephosphorylating EIIA. This unphosphorylated EIIA inhibits adenylate cyclase Cya, lowering intracellular cAMP and priming the cell for metabolic reactivation [56,57]. In the case of alanine, Tar and Trg detect the amino acid and transmit the signal via the CheA and CheY, triggering chemotactic movement. CheA also interacts with dephosphorylated EIIA, further reducing Cya activity and cAMP levels. Conversely, EIIA generated during glucose sensing can modulate CheA signaling, indicating bidirectional coupling between nutrient uptake and chemotaxis pathways. Together, these interactions coordinate metabolic reactivation with the re-engagement of motility-associated signaling, preparing cells for directed movement upon awakening [58,59].

Remarkably, awakening persister cells rapidly exhibit chemotactic movement toward nutrients [54], indicating that they retain functional sensing and motility capacity during dormancy. This behavior supports the view that persistence is driven, at least in part, by energy limitation rather than irreversible physiological shutdown [25]. Nutrient scarcity has been suggested as a primary trigger for dormancy, as insufficient energy prevents sustained protein synthesis [26]. Accordingly, the reappearance of nutrients functions as both as a metabolic stimulus and as a signal of environmental improvement, promoting the transition from dormancy to active growth [54].

## 4. Energy Status as a Driver of Dormancy Entry and Awakening

### ATP Depletion Induces Persistence, Whereas ATP and cAMP Dynamics Govern Recovery

The transition into and out of the persister cell state is tightly linked to cellular energy status. Entry into dormancy is commonly associated with a decline in intracellular ATP, whereas successful awakening requires ATP replenishment and reactivation of core metabolic pathways. Even a moderate reduction in ATP is sufficient to induce a persister-like state [25]. Because most antibiotic targets depend on ATP-driven processes, cells with reduced ATP levels show diminished target engagement and acquire transient antibiotic tolerance [27].

Under nutrient-rich conditions, exponentially growing *E. coli* cells maintain ATP concentrations of approximately 1–2 mM, whereas nutrient limitation reduces ATP levels to ~200 µM [60]. This decline suppresses protein synthesis [61], weakens chaperone-mediated folding [46], and reduces metabolic flux, collectively driving cells into a low-energy, non-growing state [25,27]. In addition to its role as an energy currency, ATP serves as a biological hydrotrope maintaining protein solubility. ATP depletion therefore destabilizes proteostasis, promoting widespread protein aggregation and formation of aggresomes—key molecular features of dormancy and antibiotic tolerance [46,49,62,63]. Residual ATP is preferentially allocated to essential ATP-dependent proteases and chaperones to preserve critical functions and support future recovery [28,60].

Consistent with this model, cells with ATP levels comparable to those of stationary-phase cultures exhibit similar antibiotic tolerance [27]. However, ATP depletion is not the sole route to persistence. Manuse et al. showed that a subset of cells with near-normal ATP levels can survive antibiotic treatment, indicating that ATP-independent mechanisms also contribute to persister formation [25].

Energetic stress in *E. coli* is sensed through the cAMP–CRP regulatory system [64]. Reduced phosphotransferase system (PTS) flux during nutrient limitation activates adenylate cyclase, increasing intracellular cAMP. The resulting CRP–cAMP complex represses growth-related gene expression while promoting stress responses and alternative carbon metabolism, reinforcing metabolic slowdown and dormancy [64].

Awakening reverses this low-energy state. Upon antibiotic removal and nutrient replenishment, persister cells actively sense improved conditions [54]. PTS-mediated nutrient uptake inhibits adenylate cyclase, lowering cAMP levels and dissociating the CRP–cAMP complex. This transcriptional switch restores growth-oriented gene expression while allowing ATP to accumulate, supporting metabolic reactivation and biosynthesis [64].

Before division can resume, persister cells must reactivate major ATP-dependent processes, including translation, DNA replication, and cell division. Cells that successfully awaken typically show a gradual increase in ATP prior to the first division, whereas cells that fail remain ATP-depleted and lyse or enter a VBNC-like state [25]. Rising ATP levels fuel DNA repair as well as DnaK–ClpB–mediated disaggregation of misfolded proteins, processes that are essential for productive regrowth [46].

Together, ATP restoration and cAMP reduction form a regulatory switch that governs the transition from dormancy to growth during persister awakening.

## 5. Ribosome Availability as a Key Modulator of Awakening Speed

### Pre-Existing and Newly Synthesized Ribosomes Determine the Success of Recovery

Ribosomes play a central role in the transition between dormancy and awakening in persister cells. During entry into the persister state, translational arrest and ribosome inactivation conserve energy by preventing protein synthesis. This occurs through the formation of 100S ribosome dimers, a hibernating form characteristic of stationary and persister cells [65,66].

Under nutrient limitation, accumulation of the alarmone (p)ppGpp triggers transcriptional reprogramming that suppresses ribosome biogenesis and activates ribosome inactivation pathways [65]. Specifically, (p)ppGpp induces expression of *rmf* (ribosome modulation factor) [67,68], *hpf* (hibernation-promoting factor) [66], and *raiA* (ribosome-associated inhibitor) [66], which together promote 70S ribosome dimerization via an inactive 90S intermediate [69] to form 100S complexes [65,70]. These dimers are stabilized during starvation and protect ribosomes from degradation, ensuring their availability for rapid reactivation once conditions improve [54,71].

The cAMP–CRP regulatory system also contributes to ribosome hibernation. Under energy stress, elevated cAMP enhances expression of *rmf* and *raiA*, reinforcing translational repression [64,66,68]. Upon nutrient replenishment, glucose or alanine uptake through the PTS system lowers cAMP levels [54], thereby relieving the translational inhibition. Ribosome reactivation is then driven by the GTPase HflX, which dissociates 100S dimers into active 70S monomers in under one minute [72,73]. The speed of this process directly influences awakening growth [65,66].

Not all cells undergo ribosome dimerization to the same extent, suggesting a threshold amount of inactive ribosomes may be required to establish or maintain the persister state [47,65]. Cells that do not reach this threshold may remain metabolically active and more susceptible to antibiotics. Conversely, excessive ribosome inactivation may delay awakening, reflecting a trade-off between protection and recovery efficiency [65].

The number of functional ribosomes at the start of recovery strongly influences lag time. Using time-lapse microscopy of rifampicin-induced *E. coli* persister cells, Kim et al. showed that cells with higher ribosome content before antibiotic treatment resumed growth immediately after drug removal, whereas those with fewer ribosomes remained dormant for hours or failed to recover [47]. Immediate awakeners contained approximately 4-fold more ribosomes than delayed ones. Cells with very low ribosome levels required de novo synthesis of rRNA and ribosomal proteins before division, resulting in prolonged lag phases [47].

Interestingly, no significant difference in ribosome content was observed between delayed awakeners and non-recovering cells [47]. However, only delayed awakeners gradually increased ribosome content during recovery, whereas non-recovering cells did not. This suggests that a minimal ribosomal threshold is required for division. Cells capable of reinitiating rRNA synthesis awakened more efficiently, underscoring the important role of ribosome biogenesis in awakening [47].

These findings show that lag time is strongly determined by ribosome availability. Persister cells that retain abundant ribosomes can rapidly resume translation by reactivating preexisting complexes with HflX [66]. In contrast, cells with depleted ribosome pools must first restore ATP levels and synthesize new ribosomes, resulting in extended lag phases. Ribosome heterogeneity established before dormancy persists into the persister state: rapid awakeners rely on ribosomes produced before antibiotic exposure, whereas delayed awakeners must rebuild their translational machinery [47].

However, ribosome abundance alone is not sufficient to guarantee successful awakening. Some cells with adequate ribosome levels still fail to recover, indicating that additional factors—proteostasis repair, metabolic reactivation, and ATP recovery—are also required [42,74,75]. Nevertheless, ribosome content remains one of the strongest predictors of awakening dynamics: the fewer ribosomes a persister retains, the longer the lag needed to rebuild translational capacity and return to growth [47].

## 6. Protein Aggregation as a Molecular Signature of Dormancy and a Barrier to Resuscitation

### Proteome Condensation Stabilizes Persistence but Requires ATP-Dependent Disaggregation for Growth Restart

As bacterial cells enter the stationary phase or face nutrient depletion, they progressively accumulate intracellular protein aggregates. These assemblies sequester essential proteins involved in transcription, translation, metabolism, and cell division [46,49,62,76,77]. Sequestration of metabolic enzymes suppresses ATP synthesis, leading to energy depletion and promoting entry into dormancy and antibiotic tolerance [25,46]. Importantly, aggregation alone is not sufficient to induce dormancy. Bollen et al. proposed that dormancy emerges only once aggregates sequester a critical threshold of energy-producing enzymes, leading to functional shutdown and protecting the cell from antibiotic-induced damage [49]. Thus, aggregate composition and load—not merely presence—determine whether aggregation drives persistence [49,50,78].

During the transition into dormancy, Yu et al. observed the formation of two compact aggregates positioned at opposite cell poles [62]. These structures, referred to as cell-pole granules, were identified by tracking fluorescently tagged FtsZ (a key bacterial cell-division protein) which shifted from a soluble cytosolic state in the exponential phase to an insoluble granule-associated state in the stationary phase. Cell-pole granules were tightly packed, associated with the inner membrane, and remained intact after cell lysis, highlighting their compact and poorly soluble nature [62]. Although the term “growth-delay bodies” was later adopted, reflecting their functional consequence: aggregates delay regrowth by storing essential proteins [62].

For dormant cells to initiate awakening, protein aggregates must be dissolved to release stored metabolic and division factors [49,50,62]. Cells that successfully dissolve aggregates exit the lag phase and resume growth, whereas those retaining aggregates remain non-growing and fail to divide [62]. Aggregate dissolution, therefore, represents an early and required checkpoint during the lag phase of awakening; failure to clear aggregates traps cells in a dormant phenotype [46,50].

Aggregate formation is progressive and heterogeneous, leading to variation in size, composition, and maturity across individual cells [50,62,79]. As starvation persists, aggregates increase in size and complexity, reflecting distinct dormancy trajectories, and generating variability in dormancy depth within the population. Larger or more mature aggregates correlate with deeper dormancy, extended lag phases, and enhanced antibiotic survival, whereas cells with fewer, more dynamic aggregates awaken rapidly and regain antibiotic sensitivity [49,50,62]. Early aggregates form liquid-like condensates that dissolve readily. Still, over time, they undergo a phase transition to rigid, solid-like assemblies stabilized by β-sheet interactions, driven in part by ATP depletion. This maturation reduces reversibility and slows disaggregation, extending the lag phase [50,80]. If aggregates become irreversibly solid, cells remain in a VBNC-like state [50].

Successful awakening thus requires dissolution of aggregates to restore protein functionality [50,62,81]. Efficient dissolution requires ATP-dependent disaggregation driven by the DnaK–ClpB machinery [82]. DnaK recognizes aggregated proteins and recruits ClpB, enabling efficient extraction and refolding [46,82,83]. ATP availability is therefore essential for awakening [82]. During recovery, ATP levels rise and DnaK–ClpB foci form at aggregates, driving disaggregation and enabling metabolic reactivation [46]. In contrast, VBNC-like cells rarely recruit DnaK and fail to dissolve aggregates (Figure 2) [46]. Intriguingly, Yu et al. reported that DnaK and ClpB localize adjacent to, but not within, compact polar aggregates, suggesting that disaggregation may begin at aggregate peripheries or require prior structural loosening, but the precise mechanism remains unclear [62].

Cells lacking *dnaK* survive antibiotics but fail to awaken efficiently, demonstrating that tolerance alone is insufficient without active disaggregation capacity [50]. Thus, protein aggregate dissolution acts as a molecular gate for awakening: only cells with functional chaperone systems and sufficient ATP can reactivate their proteome and exit dormancy [46,50].

Finally, the need to dissolve aggregates enforces a mandatory lag phase. Regrowth cannot begin until aggregated proteins are refolded and reactivated; so, aggregate resolution determines lag duration—the basis for Yu et al. naming these structures “growth-delay bodies” [62].

## 7. Antibiotic-Induced Damage Determines Recovery Trajectories

### β-Lactam and Fluoroquinolone Exposure Impose Distinct Molecular Lesions Shaping Awakening Outcomes

Antibiotic persister cells, traditionally defined as transiently tolerant cells that survive lethal exposures without acquiring heritable resistance, were long assumed to remain dormant and undamaged during treatment [4,21]. However, increasing evidence demonstrates that persister cells frequently accumulate molecular or structural lesions during antibiotic exposure, which become evident during their awakening phase [36,45]. Thus, persistence is not a purely passive escape from killing; successful awakening depends on how cells handle antibiotic-induced damage [28,36,55,84].

After antibiotic withdrawal, persister cells do not resume growth uniformly but instead follow distinct recovery trajectories determined by the nature and severity of inflicted damage. The majority of antibiotic-exposed cells are unable to reinitiate growth and are classified as dead. Among surviving cells, three recovery outcomes are commonly distinguished: healthy persisters, damaged persisters, and failed persister cells [45]. Healthy persister cells rapidly resume division, undergo symmetric cell division, produce morphologically normal progeny, and display exponential growth comparable to untreated cells (Figure 3a). Damaged persister cells exhibit temporary elongation or structural abnormalities and produce a mixture of viable and nonviable daughter cells (Figure 3b–d). Failed persister cells initiate elongation or partial growth but are ultimately unable to complete cell division and do not form colonies (Figure 3e,f). Survival alone does not guarantee regrowth; cells must also repair or redistribute accumulated damage to successfully awaken [36,45,84].

The type of damage acquired by persister cells is directly linked to the antibiotic’s mode of action. β-lactams, such as ampicillin, inhibit peptidoglycan crosslinking by targeting penicillin-binding proteins (PBPs), compromising cell wall integrity [85]. As a result, β-lactam-exposed persister cells may display triangular, branching (Figure 3c) [45] or spheroplast–like (Figure 3b) [37] morphologies indicative of defective cell wall synthesis [45]. During regrowth, these cells frequently undergo asymmetric division, known as partitioning, whereby one daughter inherits the majority of structural defects and often becomes non-growing or severely impaired, while the other daughter emerges relatively intact and resumes normal proliferation [37,45]. This strategy allows the damaged persister cell to maintain at least one viable lineage while directing accumulated lesions into a sacrificial progeny and preventing the loss of both descendants [45].

Fluoroquinolone-damaged persister cells undergo a different recovery program shaped by DNA damage. Fluoroquinolones stabilize DNA gyrase–DNA cleavage complexes, leading to stalled replication forks and the formation of single-strand or double-strand breaks (DSBs) [86]. During awakening, persister cells activate the SOS response [87], a DNA damage-induced pathway that suppresses septation and promotes DNA repair [88]. Elongation during awakening is a common phenotype (Figure 3d) and, although most pronounced following fluoroquinolone exposure, can also occur after β-lactam treatment, though less frequently and to a lesser extent [47,55]. As recovery progresses, elongated cells undergo multiple asymmetric septation events [36]. Early divisions near the cell pole often generate nonviable progeny that inherit damaged DNA, while later divisions produce viable daughter cells [36,45]. Similar to β-lactam-treated persister cells, this damage partitioning mechanism enables asymmetric segregation of lesions, facilitating the emergence of healthy progeny while sacrificing damaged lineages [45].

Thus, the nature of antibiotic-induced lesions directly determines the requirements for successful awakening. While β-lactam–exposed persister cells primarily depend on rapid detoxification to restore cell wall synthesis and resume growth, fluoroquinolone-treated persisters additionally require efficient DNA repair to resolve replication-associated lesions. These distinct damage profiles define which recovery pathways become rate-limiting during the lag phase.

## 8. Efflux-Mediated Detoxification as an Essential Requirement for Post-Antibiotic Regrowth

### Clearance of Residual Intracellular Antibiotics Limits Lag Time and Enables Cell Wall Synthesis

Following antibiotic-induced damage, persister cells must actively reduce intracellular drug levels to restore essential cellular functions. If residual antibiotic remains inside the cell, it continues to inhibit its targets. As a result, cells cannot fully reactivate metabolism or initiate division [45].

A key mechanism enabling intracellular detoxification is active efflux. In *E. coli*, the resistance-nodulation-division (RND) system AcrAB–TolC forms a tripartite pump composed of AcrB (inner membrane transporter), AcrA (periplasmic adaptor), and TolC (outer membrane channel) [89,90,91]. Powered by the proton motive force, this system exports many toxic compounds, including fluoroquinolones and β-lactams [45,90,91,92]. By lowering intracellular drug levels, efflux pumps contribute to antibiotic tolerance and are central to persister cell survival during treatment and early recovery [92,93].

Efflux activity is heterogeneous within clonal populations. Variation in *acrAB* and *tolC* expression creates subpopulations with higher efflux capacity and transiently increased tolerance [90,94]. Cells with elevated efflux accumulate less antibiotic and are more likely to survive treatment. Increased *acrAB–tolC* expression also correlates with enhanced persister formation and lower intracellular antibiotic burden during β-lactam exposure [95].

Efflux may contribute to survival during treatment by limiting intracellular drug entry. However, even cells with low basal efflux can rely on their reactivation during awakening. When nutrients become available, metabolism restarts, the proton motive force is regenerated, and efflux pumps clear residual antibiotics [45,96].

Experimental observations support the role of detoxification in shaping dynamics of awakening. In ampicillin-treated cells, higher drug concentrations produced proportionally longer lag times [45]. This suggests that the rate of intracellular antibiotic clearance determines the pace of awakening. Fang et al. proposed a positive feedback model in which initial drug efflux allows for partial metabolic reactivation, which, in turn, enhances efflux activity and accelerates recovery [45].

Although efflux is the dominant detoxification mechanism, other processes may assist. After fluoroquinolone treatment, elongated cells may dilute intracellular drug by increasing cell volume, although this remains speculative. This effect, if present, likely complements efflux activity rather than replacing it [36].

Overall, removing residual antibiotics is essential for persister cell awakening. Efflux systems such as AcrAB–TolC provide the primary route for intracellular detoxification and enable the shift from passive survival to active growth. The speed of efflux reactivation—shaped by heterogeneity in efflux gene expression and metabolic reawakening—directly influences lag duration and the probability of successful re-growth [45].

## 9. DNA Repair Pathways as Central Determinants of Fluoroquinolone Persister Recovery

### SOS-Driven Homologous Recombination and Transcription-Coupled Repair Govern Successful Resuscitation

In contrast to β-lactam exposure, fluoroquinolone treatment primarily compromises genome integrity, making DNA repair—rather than detoxification—the dominant bottleneck for post-antibiotic recovery. Persister cells that survive fluoroquinolone exposure and successfully enter awakening often carry DNA damage that must be repaired before growth can resume [87]. Although persistence was previously linked to metabolic dormancy and lack of target engagement [3,87,97], recent studies show that fluoroquinolone-exposed persister cells accumulate DNA lesions similar to antibiotic-sensitive cells [36,84]. Thus, persister dormancy does not guarantee antibiotic-induced damage-free status. The type and severity of DNA damage vary between cells, contributing to heterogeneous awakening dynamics and lag times [36].

In β-lactam-treated persister cells, there is no evidence for dedicated cell-wall repair programs. Dormant or growth-arrested cells do not synthesize peptidoglycan, and β-lactam-induced wall defects are not repaired. Instead, these cells rely on detoxification and damage partitioning, followed by normal wall synthesis once intracellular antibiotic levels drop below inhibitory thresholds [45]. By contrast, recovery from fluoroquinolone exposure requires repair of DNA double-strand breaks (DSBs) generated by stabilized cleavage complexes formed during inhibition of DNA gyrase or topoisomerase IV [86,98]. Both persister and sensitive cells accumulate comparable levels of DNA damage under ofloxacin treatment, and DNA repair efficiency strongly affects awakening success and timing [28,36,84,99].

During early recovery, fluoroquinolone-treated persister cells activate the SOS response, a global regulatory system triggered by RecA-mediated sensing of single-stranded DNA and controlled by the LexA repressor [87]. SOS induction suppresses cell division and promotes DNA repair [88]. In persister cells SOS activation peaks early, often accompanied by filamentation with multiple segregating nucleoids [84]. Division resumes only after genomic integrity is restored. Impairment of SOS-dependent repair significantly reduces survival, as demonstrated by the finding that only 21% of wild-type filaments complete division, compared to approximately 5% in SOS-deficient mutants, underscoring the essential role of SOS-regulated processes in persister recovery [36].

Double-strand break repair is mainly mediated by RecBCD-dependent homologous recombination (HR) in *E. coli*. Key HR proteins include RecA, RecB, RecC, RuvA/B, and RecG, many of which are induced under SOS control [100,101]. Efficient DSB repair requires a homologous template; so, cells with two chromosome copies before treatment show dramatically higher survival: diploid-enriched populations persist ~40-fold better than monoploid ones [84,102]. Monoploid persister cells may survive only if damage is limited or if alternative repair pathways act. Thus, chromosome copy number and timely homologous recombination are major determinants of awakening success after FQs treatment [84,103].

Fluoroquinolones also induce oxidative DNA lesions that stall RNA polymerase. Stalled polymerase serves as a signal for DNA damage and recruits the transcription-coupled repair (TCR) machinery [44,104,105]. During recovery in fresh medium, transcription re-initiates, enabling TCR to detect and repair oxidative DNA lesions via nucleotide excision repair (NER) [104]. Stalled RNA polymerase is cleared through two alternative TCR pathways: UvrD-mediated backtracking [106] or Mfd-mediated forward displacement (Figure 4) [107].

In the UvrD pathway, RNA polymerase backtracks but remains associated with DNA, allowing for rapid resumption of transcription after NER-mediated repair [106]. In contrast, the Mfd pathway displaces RNA polymerase forward, terminates transcription, and requires a new initiation event, resulting in longer awakening times [107,108]. UvrD-mediated TCR is associated with shorter lag phases, whereas Mfd-mediated TCR extends lag duration [44]. Mfd functions as an evolvability factor, and a subset of persister cells using this pathway generates mutations that may promote the emergence of resistance [44,109].

These competing TCR pathways generate heterogeneous awakening kinetics [44]. Importantly, premature growth before repair completion is lethal: delaying DNA repair until growth restarts sharply reduces survival [103]. Successful persister cells delay division until repair is complete [84].

Some persister cells fail to complete repair and enter an abortive recovery state. These cells may elongate initially but cannot divide [36,45]. Inadequate DSB repair, unresolved oxidative lesions, or insufficient antibiotic detoxification prevent restoration of cellular homeostasis and preclude productive regrowth. β-lactam-exposed persister cells primarily rely on detoxification and damage partitioning. In contrast, fluoroquinolone-exposed persister cells must first engage SOS-controlled recombination and excision repair, potentially aided by dilution of damage during filamentation. Only after successful completion of these processes can persister cells initiate productive division, often via asymmetric segregation of lesions into nonviable progeny, thereby preserving a healthy lineage [45].

## 10. Induction Pathways Impose Indirect Constraints on Persister Awakening

### Stringent Response and Toxin–Antitoxin Systems Condition, but Do Not Dictate, Lag-Phase Duration

Persister cells can arise through distinct induction mechanisms, including global metabolic downregulation mediated by the stringent response and growth arrest enforced by toxin–antitoxin (TA) systems. Although these pathways differ in their molecular implementation, both promote dormancy by suppressing core growth processes and impose similar constraints on post-antibiotic recovery, in that their activity must be reversed before awakening can proceed [103].

The stringent response is mediated by accumulation of the alarmone (p)ppGpp during nutrient or energy stress. Elevated (p)ppGpp suppresses ribosome biogenesis, translation, and growth-associated transcription, thereby promoting entry into dormancy and persister formation [110,111]. Sudden increases in (p)ppGpp rapidly inhibit growth and protein synthesis, whereas basal (p)ppGpp levels modulate ribosome abundance and growth rate under non-stress conditions [111]. Despite its central role in persister induction, the stringent response does not appear to actively instruct awakening or define lag-phase duration. Instead, reduction in intracellular (p)ppGpp during recovery is permissive for growth resumption, allowing awakening to proceed only after downstream processes such as ATP restoration, ribosome reactivation, and proteostasis recovery have progressed [111,112].

Available evidence indicates that the decline in (p)ppGpp during recovery is not driven by a dedicated awakening signal but reflects reduced synthesis combined with growth-dependent dilution once stress conditions are relieved. In experimental systems inducing the stringent response with serine hydroxamate (SHX), *E. coli* gradually recovers from growth arrest as intracellular (p)ppGpp levels decrease to pre-stress concentrations over several hours [112]. Thus, while (p)ppGpp levels must ultimately fall to enable full metabolic reactivation and growth resumption, their decrease does not actively set awakening timing.

Importantly, although (p)ppGpp does not directly regulate awakening kinetics, it participates in DNA damage processing during early recovery. (p)ppGpp binds RNA polymerase and promotes transcriptional backtracking at sites of DNA lesions, facilitating lesion recognition and removal by nucleotide excision repair (NER) machinery [113,114]. This mechanism parallels the UvrD-mediated backtracking pathway described in Section 9, highlighting functional coupling between transcriptional regulation and DNA repair during recovery from fluoroquinolone-induced damage. In addition, (p)ppGpp modulates components of the SOS response and homologous recombination pathways, supporting genome integrity under conditions of replication stress and DNA strand breaks [115]. Thus, maintenance of (p)ppGpp during early recovery may indirectly benefit cells by allowing DNA repair to proceed before complete exit from dormancy, even though (p)ppGpp itself does not dictate lag-phase length.

In contrast, TA systems induce persistence through direct inhibition of essential cellular processes [116]. Type II TA modules such as RelBE and MazEF encode stable protein toxins (RelE and MazF) and labile protein antitoxins (RelB and MazE) within the same operon [117,118]. Under stress, preferential degradation of the antitoxin by cellular proteases releases the toxin, leading to reversible bacteriostatic arrest through inhibition of translation or replication [116,119]. Direct experimental evidence for TA-mediated effects on recovery comes from studies using MazF-induced persisters. Cells arrested by MazF display tolerance to both β-lactams and fluoroquinolones and show delayed resumption of growth and DNA synthesis after ofloxacin treatment [103]. In this system, recovery requires de novo synthesis of the cognate antitoxin MazE to neutralize MazF and relieve inhibition of macromolecular synthesis [103].

Because antitoxins are intrinsically unstable, their accumulation requires partial restoration of transcription and translation, introducing an inherent delay before growth-associated processes can resume. In fluoroquinolone-treated MazF persisters, this delay coincides with the time required for DNA repair, providing a temporal window during which fluoroquinolone-induced lesions can be resolved before replication restart renders unrepaired damage lethal [103]. Consistent with this model, ectopic expression of MazE accelerates growth resumption but reduces the fraction of cells completing successful recovery, consistent with premature awakening before DNA repair is complete [103]

At the population level, deletion of multiple TA systems does not significantly alter awakening heterogeneity or lag-time distributions. An *E. coli* strain lacking ten TA modules (Δ10TA) displays regrowth dynamics indistinguishable from wild-type cells, indicating that TA systems are not universal determinants of persister awakening kinetics [34]. Their influence becomes apparent primarily when persistence is generated through toxin activation rather than global metabolic downregulation [103].

Thus, both (p)ppGpp accumulation and toxin activity can persist into early recovery for mechanistic reasons—DNA repair engagement in the former case and delayed antitoxin replenishment in the latter—indirectly extending the lag phase without acting as direct regulators of awakening timing.

## 11. Conclusions

### Persister Cell Survival Becomes Clinically Relevant Only When It Is Followed by Successful Physiological Reactivation and the Formation of Viable Progeny

Antibiotic persistence is not a static dormant state but a dynamic process that involves survival, physiological reactivation, and restoration of reproductive capacity. While persister cells tolerate antibiotic exposure, they are often not damage-free. Depending on the drug’s mode of action, surviving cells may carry cell wall defects [45] or molecular lesions such as DNA breaks [36,84] or oxidative damage [44]. As a result, awakening is heterogeneous and depends on each cell’s metabolic, structural, and molecular state at the time of dormancy entry.

Following antibiotic removal, persister cells enter a lag phase, a critical window during which they must reverse dormancy-associated changes and process antibiotic-induced damage. Lag duration varies widely and is shaped by dormancy depth, reflecting the integrated effects of ATP availability, ribosome content, and proteostasis status, and the type and severity of damage. These intrinsic and damage-related determinants of awakening are summarized in Table 2. Cells with sufficient ATP, functional ribosomes, and low aggregate burden awaken rapidly. Deeply dormant cells with extensive aggregation, low ATP, or severe DNA lesions require extended recovery. Some fail to restore homeostasis and transition into a failed persister state or a viable-but-nonculturable (VBNC) condition, unable to generate progeny. Upstream mechanisms such as (p)ppGpp accumulation or toxin activation primarily shape dormancy depth, whereas lag-phase duration is determined by the extent of downstream repair and metabolic reactivation required for growth resumption.

Successful awakening requires coordinated reactivation of metabolic and biosynthetic pathways (Figure 5). ATP regeneration supports repair and remodeling processes, including DnaK-ClpB-mediated dissolution of protein aggregates [46,50,62]. Ribosome reactivation via dissociation of 100S dimers allows translation to restart; persister cells with low ribosome abundance must synthesize new ones before division, prolonging lag time [47,65]. Persister cells unable to restore proteostasis remain metabolically trapped and do not resume division [50].

Antibiotic-induced damage further shapes recovery. After β-lactam exposure, persister cells must clear residual intracellular drug via efflux before peptidoglycan synthesis resumes [45]. After exposure to fluoroquinolones, double-strand and oxidative DNA lesions require activation of SOS-regulated homologous recombination and nucleotide excision repair [36,84]. Filamentation may help dilute and segregate damage [36,45]. Many persister cells divide asymmetrically, directing damage to a non-viable daughter while preserving a healthy lineage [45].

Importantly, the mode of persister induction can impose additional temporal constraints on awakening. In persisters generated through toxin–antitoxin system activation, the requirement for de novo antitoxin synthesis introduces a coincidental delay that can be beneficial under conditions requiring extensive damage repair, such as after fluoroquinolone exposure, thereby indirectly increasing the probability of successful recovery [103].

Not all persister cells complete recovery. Cells that cannot detoxify antibiotics, repair DNA, or restore proteostasis fail during the lag phase. Once a persister cell completes the first division, its progeny typically regain exponential growth and drug sensitivity [45,47].

The duration of the lag phase during persister awakening is shaped not only by intracellular processes but also by extracellular factors, particularly the structural context in which cells reside. Planktonic cells, which experience relatively uniform nutrient availability and efficient diffusion of antibiotics and metabolites, typically display faster and more homogeneous awakening kinetics in general. In contrast, the biofilm-associated lifestyle profoundly alters both persister formation and the dynamics of post-antibiotic recovery [120,121]. Biofilms create steep nutrient and oxygen gradients that reduce metabolic activity and ATP availability, conditions strongly associated with elevated persister frequencies in *E. coli*. Within this structured community, impaired proton motive force and slower ribosome activation further delay exit from dormancy, contributing to extended lag phases following antibiotic removal [3,4]. Additionally, protein aggregates—whose dissolution is required for awakening—persist longer in biofilm-embedded cells due to limited energy resources and reduced chaperone activity [5].

A growing body of evidence indicates that core features of persister awakening identified in *E. coli*—ATP restoration, recovery of ribosomal function, and repair of antibiotic-induced damage—are broadly conserved across clinically relevant pathogens [3,28,122]. However, species-specific physiology can markedly reshape awakening dynamics. In *M. tuberculosis*, extremely slow growth, a lipid-rich and impermeable cell envelope, and reliance on respiratory metabolism contribute to prolonged and heterogeneous lag phases [123,124]. *Pseudomonas aeruginosa* shows awakening strongly influenced by its high efflux capacity, redox flexibility, and biofilm-associated physiology [125,126,127]. In *Staphylococcus aureus*, deep ATP depletion, formation of non-proliferating SCVs, and the highly cross-linked Gram-positive cell wall slow antibiotic penetration and delay recovery, making awakening dependent on respiration-driven ATP restoration and autolysin reactivation [128,129,130].

In summary, persistence is best understood as a multistep process comprising survival during antibiotic exposure, effective detoxification and/or repair during the recovery phase, and eventual restoration of growth capacity. It represents transient antibiotic tolerance followed by a damage-processing period that determines whether a cell can restore its physiological integrity and resume growth. The lag phase is the decisive window in which survival is converted into reproductive viability or lost. The ability to reestablish energy balance, reawaken the translational machinery, restore proteostasis, and repair molecular lesions determines whether a surviving cell resumes growth or remains trapped in dormancy. Understanding these recovery processes reveals key vulnerabilities that could be targeted to prevent bacterial resurgence after antibiotic treatment.

## 12. Future Perspectives and Clinical Relevance

Despite substantial progress in understanding the molecular mechanisms that govern persister cell awakening, several key questions remain. Future research should clarify how metabolic reactivation, proteostasis restoration, and DNA repair are temporally coordinated within individual cells, and whether disrupting this synchronization could selectively eliminate persister cells during the lag phase. Likewise, defining critical molecular thresholds—such as minimal ATP levels, ribosome content, and disaggregation capacity—may identify vulnerable transition points that can be pharmacologically exploited.

From a clinical perspective, strategies that target the lag phase represent a conceptual opportunity to combat recurrent infections. Approaches that prevent intracellular detoxification, inhibit DNA repair pathways, block chaperone-mediated aggregate dissolution, or prematurely force cell-cycle re-entry may convert awakening persister cells into antibiotic-susceptible targets. Conversely, strategies that lock persister cells in an irreversible dormant state may prevent relapse by blocking their ability to resume growth.

These mechanisms are likely to be particularly relevant in infections characterized by repeated antibiotic exposure and fluctuating nutrient conditions, such as urinary tract infections, tuberculosis, and cystic fibrosis–associated lung disease [12,13,14,15,16,17]. In these settings, heterogeneous metabolic states, biofilm growth, and intracellular niches are expected to prolong lag phases and favor persister survival. Molecular features associated with delayed awakening—such as low ATP levels, elevated protein aggregation, altered ribosome content, or sustained SOS activation—may therefore serve as potential biomarkers of persister-enriched populations, although their clinical validation remains an open challenge. At present, these features should be viewed as exploratory markers rather than clinically actionable diagnostics [25,27,36,46,47,62,103].

At the same time, translation of lag-phase–targeting strategies into clinical practice faces substantial obstacles. Drug penetration into biofilms or intracellular compartments, host toxicity of agents interfering with essential stress responses, and the physiological heterogeneity of bacterial populations in vivo may limit the efficacy of such approaches [19,131,132]. Moreover, premature forcing of growth or inhibition of repair pathways may carry the risk of selecting for resistance or causing unintended tissue damage [22]. These challenges underscore the need for careful evaluation of persister-targeting strategies in physiologically relevant infection models.

Thus, a deeper understanding of the molecular logic governing persister cell recovery strengthens the conceptual framework of persistence and may inform future therapeutic avenues for eradicating chronic and recurrent bacterial infections.

## Figures and Tables

**Figure 1 ijms-27-00467-f001:**
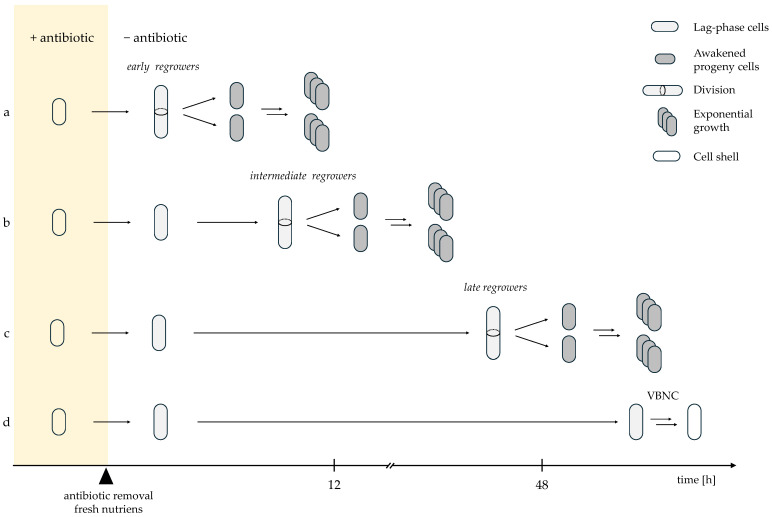
Heterogeneous awakening trajectories of persister cells following antibiotic removal; (**a**) early regrowers with rapid resumption of growth; (**b**) intermediate regrowers with moderate lag; (**c**) late regrowers requiring prolonged recovery, and (**d**) cells that fail to awaken and enter a non-dividing VBNC/failed persister state [37,47,48]. Solid arrows indicate the direction of sequential events following antibiotic removal; arrow length reflects relative differences in timing along the time axis. Double arrows denote concurrent and repetitive growth-associated processes (cell growth, DNA replication, and division), resulting in population expansion. A break in the time axis indicates a prolonged phase that is shortened for graphical clarity.

**Figure 2 ijms-27-00467-f002:**
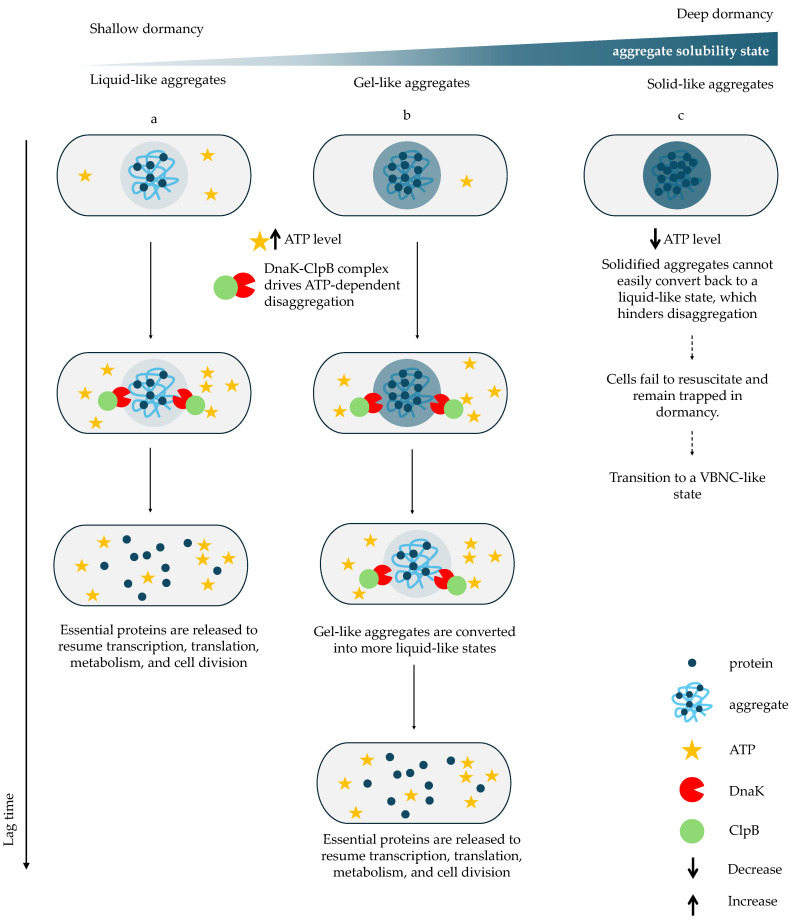
The impact of aggregate state on disaggregation and awakening kinetics. During resuscitation, ATP levels increase and the DnaK–ClpB complex assembles to dissolve aggregates and release essential proteins required to resume transcription, translation, metabolism, and cell division. (**a**) In cells with liquid-like aggregates, rapid ATP recovery enables fast DnaK–ClpB recruitment and efficient aggregate dissolution, resulting in a short lag phase. (**b**) In cells with gel-like aggregates DnaK–ClpB must first remodel the aggregates into a liquid-like state before disaggregating them, thereby delaying protein release and prolonging the lag phase. In contrast, (**c**) cells containing solid-like aggregates fail to accumulate ATP sufficiently to activate DnaK–ClpB; aggregates remain insoluble, essential proteins remain sequestered, and cells remain trapped in dormancy or enter a VBNC-like state [46,49,50]. Solid arrows indicate sequential stages along a relative vertical time axis, whereas dashed arrows denote a time-independent alternative fate leading to VBNC-like states or cell death.

**Figure 3 ijms-27-00467-f003:**
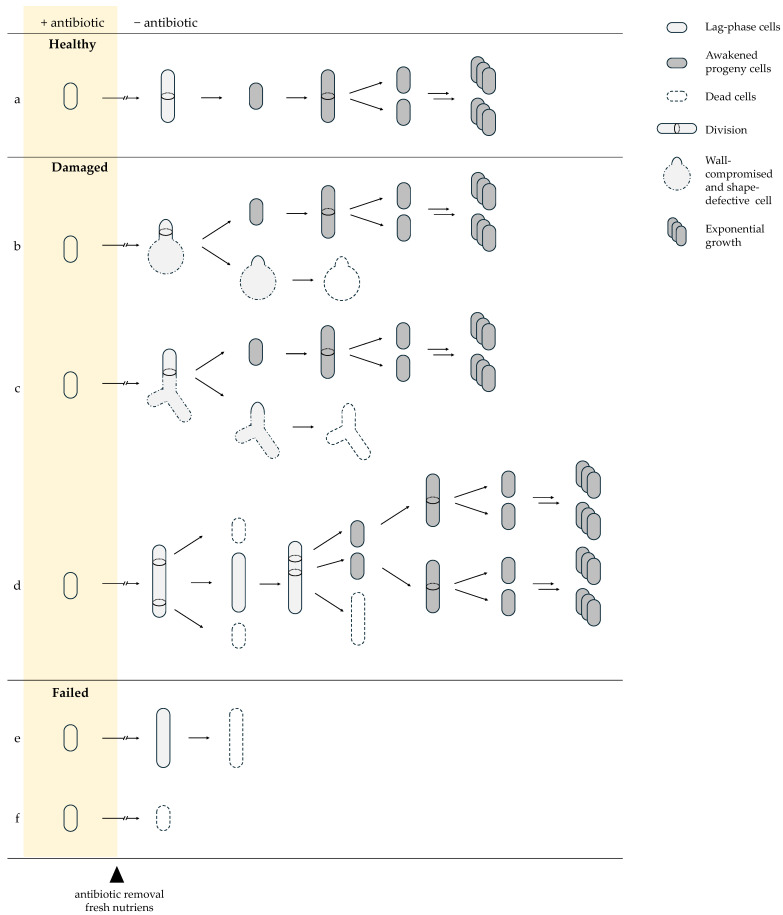
Recovery trajectories of persister cells after antibiotic removal. Surviving cells follow three principal trajectories: Healthy persisters (**a**) resume growth immediately and divide symmetrically. Damaged persisters show transient morphological defects: (**b**) spherical-like shapes [37] and (**c**) triangular/branched forms from β-lactam–induced cell-wall damage with asymmetric survival of progeny [45]; (**d**) or fluoroquinolone-induced filamentation with mixed viable and non-viable offspring due to DNA repair and segregation errors [36,45]. Failed persister cells (**e**) initiate elongation but lyse, or (**f**) collapse immediately, reflecting irreversible damage. Solid arrows indicate the direction of sequential events following antibiotic removal (arrow length is not time-scaled). Double arrows denote concurrent and repetitive growth-associated processes (cell growth, DNA replication, and division). Arrows with breaks indicate a variable, cell-specific lag phase preceding growth, differing among trajectories (**a**–**f**).

**Figure 4 ijms-27-00467-f004:**
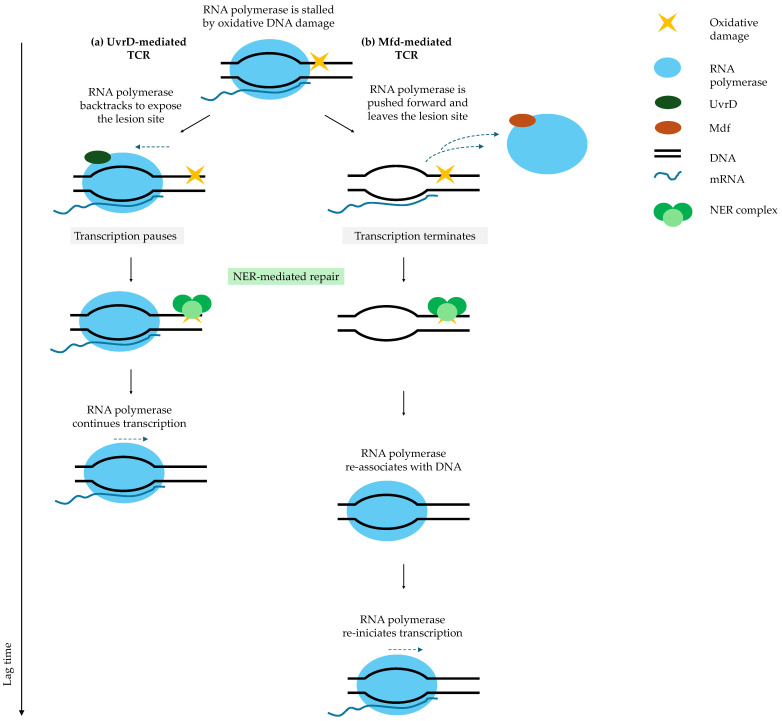
Transcription-coupled repair (TCR) pathways determine lag time after fluoroquinolone treatment. When RNA polymerase stalls at a DNA lesion, TCR can follow two pathways: (**a**) UvrD-mediated TCR: RNA polymerase backtracks to expose the lesion site, allowing nucleotide excision repair (NER) to repair the DNA. After repair, RNA polymerase molecule promptly resumes transcription. (**b**) Mfd-mediated TCR: Mfd displaces RNA polymerase forward, terminating transcription. After NER repairs the lesion, RNA polymerase must rebind DNA and re-initiate transcription, significantly prolonging the lag phase and increasing mutagenesis [43]. Solid arrows indicate the direction of sequential events following antibiotic removal (arrow length is not time-scaled). The vertical axis represents a relative timeline, illustrating that a greater number of steps corresponds to a longer overall process. Blue dashed arrows indicate alternative fates of RNA polymerase, including backtracking or dissociation from DNA.

**Figure 5 ijms-27-00467-f005:**
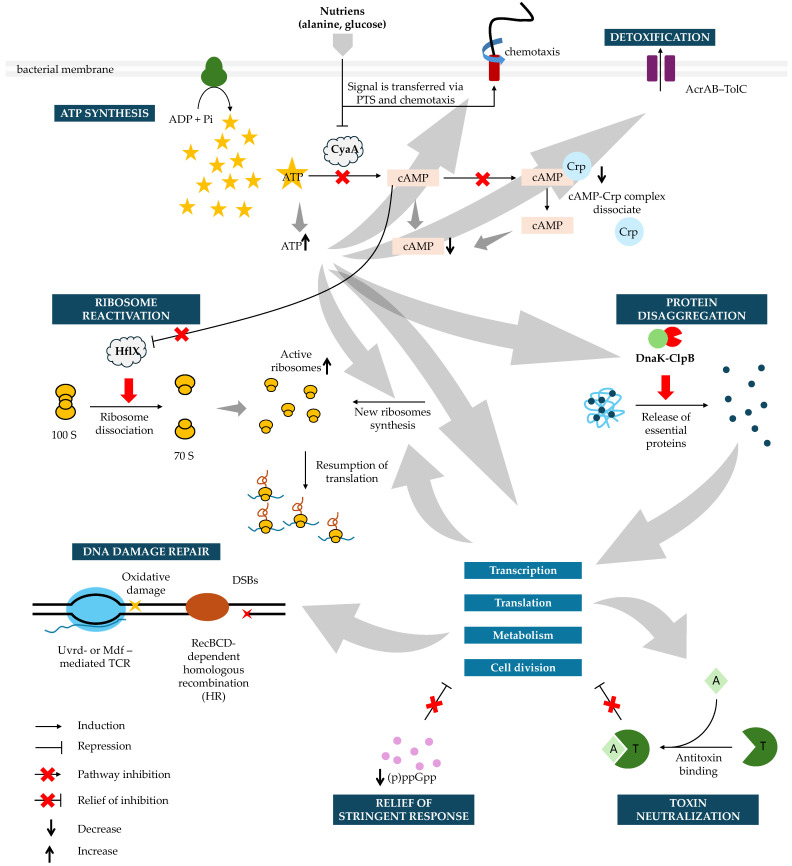
Molecular Pathways Driving Persister Cell Awakening. During the lag phase, persister cells sequentially reactivate key cellular processes to transition from dormancy to growth. Awakening begins with nutrient sensing via the phosphotransferase system (PTS) and chemotaxis systems, which lower cAMP levels, inactivate adenylate cyclase (CyaA), and allow ATP to accumulate. As recovery progresses, the stringent response is relieved through a gradual decline in intracellular (p)ppGpp levels, primarily driven by reduced synthesis and growth-dependent dilution, thereby permitting reactivation of growth-associated transcription and translation. In parallel, toxin–antitoxin–mediated growth arrest is relieved by de novo synthesis of antitoxins, which neutralize cognate toxins and release inhibition of the cell cycle, transcription, translation, and metabolism. Dissociation of the cAMP-Crp complex restores growth-directed transcription while increasing ATP fuels ribosome reactivation by HflX, and supports de novo ribosome synthesis when needed. ATP also powers DnaK-ClpB-mediated disaggregation, releasing essential proteins required for transcription, translation, metabolism, and cell division. Concurrently, efflux pumps clear residual antibiotics, enabling recovery of cell wall synthesis after β-lactam exposure. Following fluoroquinolone treatment, DNA damage is repaired by RecBCD-mediated homologous recombination (HR) and transcription-coupled repair (TCR) via UvrD or Mfd. As proteostasis, metabolism, and genome integrity are restored, cells resume transcription, translation, and growth, ultimately re-entering the division cycle and transitioning from the lag phase into exponential growth. Crp—cAMP receptor protein; DSBs—double strands breaks. Grey arrows represent general downstream influences between pathways during persister awakening and do not denote direct biochemical reactions.

**Table 1 ijms-27-00467-t001:** Glossary of key terms used in this review.

Term	Definition
Surviving cells/ survivors	Cells that remain viable after antibiotic exposure, without necessarily fulfilling the definition of persister cells; may follow healthy, damaged, failed, or VBNC trajectories.
Persister cells	A subpopulation of survivors that tolerate antibiotics without genetic resistance, enter a lag phase, successfully resume growth, and produce viable progeny.
Failed persister cells	Survivors that initiate recovery (e.g., elongation or filamentation) but fail to complete the first division and do not form colonies.
VBNC (viable but non-culturable) cells	Cells that retain membrane integrity and minimal metabolic activity but are unable to resume growth; many are structurally compromised “cell shells” and effectively non-recoverable.
Awakening	The transition of persister cells from dormancy to the first successful division, involving metabolic reactivation, detoxification, repair, and restoration of proteostasis.
Lag phase (awakening lag)	The time period between the transfer of persister cells into antibiotic-free nutrient-rich conditions and the completion of the first division.

**Table 2 ijms-27-00467-t002:** Intrinsic and damage-related factors influencing lag time in persister-cell awakening.

Category	Factor	Effect on Lag Time/ Awakening (Evidence Level) *	General Role	Ref **
Intrinsic (pre-existing before antibiotic exposure)	ATP level	Low ATP → long lag; ATP rise required for growth (S2)	ATP fuels repair, disaggregation, translation restart	[25,46,54]
	Ribosome abundance	High → fast; low → delayed (S2)	Pre-existing ribosomes enable immediate translation; low pool requires de novo synthesis	[47,65]
	Protein aggregates (load & solubility)	Liquid/small → fast; solid/large → long lag/VBNC (S2)	Aggregates must be dissolved to release essential proteins	[50,62]
	Chaperone capacity (DnaK-ClpB)	Strong → efficient awakening; weak → failure (S1)	ATP-dependent disaggregation and refolding of aggregated proteins	[46,50,62]
	Nutrient-sensing systems (PTS, chemotaxis)	Efficient sensing → fast exit from dormancy (S2)	Detect nutrients, lower cAMP, activate growth programs	[54]
	Dormancy depth	Shallow → short lag; deep → long lag/VBNC (S2)	Defines how much reactivation and repair is required	[34]
	(p)ppGpp level/ stringent response	High (p)ppGpp → deeper dormancy; indirect extension of lag (C1)	Sets dormancy depth via global translational repression	[112]
	Toxin–antitoxin–mediated growth arrest (e.g., MazEF, RelBE)	TA-induced persisters → prolonged lag (C1)	Reversible inhibition of translation and/or replication;	[103]
	Chromosome copy number (ploidy)	≥2 chromosomes → higher survival, faster recovery (S1)	Extra template improves DNA damage repair	[84]
Damage-related (acquired during antibiotic exposure)	Residual intracellular antibiotic	High concentration → long lag (S2)	Drug must be cleared before growth resumes	[45]
	Efflux activity (AcrAB–TolC)	High → short lag; low → prolonged lag (S1)	Pumps out antibiotics, enables metabolic restart	[45]
	Cell-wall damage (β-lactams)	Structural defects slow recovery (S2)	Wall rebuilt after detox; damage partitioned during division	[45]
	DNA double-strand breaks (fluoroquinolones)	More breaks → longer lag; severe → failure (S1)	Must be repaired before replication and division	[36]
	Homologous recombination (HR) RecA/RecBCD	Efficient HR → successful recovery (S1)	Repairs DSBs using intact DNA template	[84]
	Transcription-coupled repair (TCR) UvrD vs. Mfd	UvrD → shorter lag; Mfd → longer lag + mutagenesis (S1)	Repairs transcription-blocking lesions	[44]
	Filamentation and damage partitioning	Extends lag; supports survival of one lineage (S2)	Dilutes damage; asymmetric divisions	[45]

* Evidence annotation combines two complementary dimensions: Strength of evidence: S—strong causal support; C—conditional or context-dependent. Primary evidence type: 1—genetic perturbation (knockout or controlled expression); 2—single-cell approaches including microfluidics and time-lapse microscopy. In many cases, conclusions are supported by multiple experimental approaches; the dominant or most informative evidence type is indicated for clarity. ** Reference numbers indicate the primary literature supporting each factor. PTS—phosphotransferase system; DSBs—Double Strands Breaks.

## Data Availability

No new data were created or analyzed in this study. Data sharing is not applicable to this article.

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
