# Peer review of "Molecular Basis of Persister Awakening and Lag-Phase Recovery in *Escherichia coli* After Antibiotic Exposure"

_ijms, 2026, doi:10.3390/ijms27010467_

Round 1
Reviewer 1 Report
Comments and Suggestions for Authors
This is an excellent, timely, and insightful review that successfully consolidates a rapidly advancing field. The authors move beyond the traditional focus on how persister cells form and effectively spotlight the critical, yet underappreciated, phase of awakening and recovery. The writing is clear, the logic is well-structured, and the figures are helpful for visualizing complex concepts. It will be a valuable resource for researchers in the field of antibiotic persistence. However, the author can find minor comments in the attached PDF for further improvement.

Reviewer 2 Report
Comments and Suggestions for Authors
Dear Authors,
Thank you for the opportunity to review your manuscript. This is a very interesting and impactful piece of work, particularly from a clinical perspective, as bacterial persistence is strongly associated with therapeutic failure and recurrent infections.
Overall, the manuscript provides a comprehensive, up-to-date, and well-structured review of the molecular mechanisms underlying bacterial persistence in E. coli, as well as the factors that drive reactivation after antibiotic exposure. The narrative is clear, the depth of analysis is impressive, and the content is well supported by modern primary literature. However, I have several comments that I believe may help strengthen the manuscript.
-
Although the review is detailed and highly informative, several sections are exceedingly long. For example, the discussion on ATP and cAMP is well supported but contains conceptual redundancies—particularly regarding ATP as a hydrotrope and its role in chaperone-mediated repair—that could be consolidated into a single paragraph. The manuscript would benefit from a careful revision to reduce repetition and streamline key ideas for improved readability and flow.
-
Despite being a narrative review, it would be valuable to include information on the literature search strategy, such as:
– criteria for article selection
– databases consulted
– search dates
– inclusion and exclusion criteria
This addition enhances transparency and rigor, especially for a review intended for a broad scientific audience. -
The figures are conceptually strong and visually informative, but some require higher resolution, particularly Figures 1 and 2. It is also important to specify in the figure legends whether these illustrations were created by the authors or adapted from other sources.
As a suggestion, if the figures were generated in RStudio, increasing the output resolution to 300–600 dpi could significantly improve clarity without extensive adjustments. -
The conclusion and final discussion briefly address the clinical relevance of persistence, but this section could be expanded. For example:
– How do these mechanisms manifest in real infections such as UTIs, tuberculosis, or cystic fibrosis?
– Which biomarkers could potentially be used to detect persister cells?
– What therapeutic opportunities arise from targeting the lag phase?
These topics are mentioned but could be further developed to enhance translational relevance. -
Several important concepts are underdeveloped or missing.
For instance, the relationship between persister cells and biofilms deserves deeper discussion, particularly how the biofilm microenvironment modulates ATP levels, protein aggregate dissolution, ribosome activity, and lag-phase dynamics. Similarly, toxin–antitoxin systems are only briefly mentioned, but they represent a crucial mechanism in the field and merit further attention. -
The manuscript briefly distinguishes heteroresistance from persistence, but I recommend expanding this explanation. Although these terms have been known for years, the clinical relevance of heteroresistance has only recently gained attention, and a clearer distinction would greatly benefit readers.
-
In my opinion, this is a valuable and well-constructed manuscript with strong potential for publication. I hope the comments above are helpful and contribute positively to the improvement of your work.
Reviewer 3 Report
Comments and Suggestions for Authors
Reviewer report :
This is a solid and timely review on persister awakening and lag-phase recovery in E. coli. The manuscript reads well overall and pulls together a lot of recent single-cell and molecular work. The structure is clear and the figures/tables help the reader. Still, a few areas need tightening before acceptance.
Major comments
1. Right now some mechanistic statements sound like they’re equally supported by in vitro assays, single-cell imaging, and mutant studies, but that’s not always the case. Please flag clearly when findings come from microfluidic singlecell work vs. bulk culture vs. genetic knockout studies. This helps readers judge the strenght of each conclusion.
2. The clinical translation parts are interesting, but a bit speculative. Maybe add 1–2 sentences acknowledging the challenges (drug delivery, host toxicity, biofilm complexity, etc.). Not a big rewrite, just a bit more balanced so readers don't over-interpret.
3. A few 2024–2025 papers on persister imaging and mutagenesis could be added. Not many, just check small gaps esp. around TCR/Mfd-related awakening mechanisms.
4. Table 2 is helpful but maybe too long/repetitive. You could condense similar mechanisms and add a quick “evidence level” label (e.g., strong genetic, single-cell only, speculative). Figures are fine, but maybe add short captions in panels pointing to the key primary refs.
5. Please define early and use consistently: persister vs survivors vs VBNC.
There are a few truncated/odd sentences (e.g., line ~549 “formation of viable.”), probably just typos. A quick proofread will fix these.
Check gene/protein formatting (italic for genes, not proteins).
Minor
Add abbreviation legend in Table 1 (PTS, CRP, HR).
Round 2
Reviewer 2 Report
Comments and Suggestions for Authors
I thank the authors for considering my comments. I am satisfied with the changes made.
Reviewer 3 Report
Comments and Suggestions for Authors
Accept in present form